

# Hydrometric measurements in peatland-dominated, discontinuous permafrost at Scotty Creek, Northwest Territories, Canada - Changing Cold Regions Network (CCRN) Special Observation and Analysis Period (SOAP)

Kristine M. Haynes, Ryan F. Connon, William L. Quinton

Cold Regions Research Centre, Wilfrid Laurier University, Waterloo, N2L 3C5, Canada

*Correspondence to*: Kristine M. Haynes (khaynes@wlu.ca)

**Abstract.** The discontinuous permafrost region of northwestern Canada is experiencing rapid warming resulting in dramatic land cover change from forested permafrost terrain to treeless wetlands. Extensive research has been conducted throughout this region to gain insight into how climate-induced land cover change will impact water resources and ecosystem function. This paper presents a hydrological and micrometeorological dataset collected in the Scotty Creek basin, Northwest Territories, Canada over the course of the Changing Cold Regions Network (CCRN) Special Observation and Analysis Period (SOAP) year of 01 October 2014 to 30 September 2015. Micrometeorological data collected from four stations located in land cover types representative of those comprising the Scotty Creek basin, including bog, channel fen, stable peat plateau and peat plateau undergoing rapid permafrost degradation and loss are presented. Monitored micrometeorological variables include incoming and outgoing shortwave and longwave radiation, air temperature, relative humidity, wind speed, precipitation (rain and snow) and snow depth. Deep ground temperatures (~ 1 to 10 m below the ground surface) from a channel fen as well as disturbed sites common to the basin including a seismic line and winter road are presented. Water levels were also monitored in the representative land cover types over this period. This dataset is available from the Wilfrid Laurier University Library Research Data Repository (https://doi.org/10.5683/SP/OQDRJG) and can be used in coordination with other hydrological and micrometeorological datasets, including those from the CCRN, to examine spatio-temporal effects of meteorological conditions on local hydrological responses across cold regions.

## 1 Introduction

The permafrost landscape across northwestern Canada is undergoing dramatic land cover change as a result of rapid climate warming, particularly along the southern boundary of the discontinuous permafrost region (Jorgenson and Osterkamp, 2005; Quinton et al., 2011). Warming air temperatures threaten both the areal coverage and thickness of permafrost bodies via both lateral and vertical thaw in the discontinuous and sporadic permafrost zones (McClymont et al., 2013; Connon et al.,



2018). The thickness of the layer of ground above permafrost (*i.e.* supra-permafrost layer) is expected to increase as a result of warming (Frampton & Destouni, 2015; Lawrence et al., 2012), catalyzing the transition to thermokarst landscapes (Kokelj and Jorgenson, 2013). The supra-permafrost layer includes the active layer, which freezes and thaws annually, but may also include a perennially-unfrozen layer (*i.e.* talik) between the active layer and underlying permafrost (Muller, 1947). A talik

develops when the depth of thaw during summer exceeds the depth of re-freeze during winter, a process that can be initiated by an increase in air temperature, soil moisture, snow depth or other change increasing the heat flux into the ground (Åkerman and Johansson, 2008; Atchley et al., 2016). A talik accelerates permafrost thaw since it bounds the upper surface of permafrost (*i.e.* permafrost table) with a layer that is at or above the freezing point, and therefore eliminates the loss of energy from the permafrost during winter. As such, the formation of taliks may be instrumental in influencing the rate of

permafrost thaw and altering the thermal and moisture regimes across the evolving discontinuous permafrost landscape (Connon et al., 2018).

The considerable observed land cover changes resulting in the loss of permafrost-cored peat plateaus (e.g. Beilman and Robinson, 2003; Pohl et al., 2009) significantly impact the cycling of water through these landscapes (Connon et al., 2014).

Long-term increases in streamflow have been documented throughout the Northwest Territories (NWT), with the greatest increases in flow observed in the south-central NWT (St. Jacques and Sauchyn, 2009). In peatland terrains, the relative contributions of runoff sources have changed as the area contributing to basin runoff expands and previously isolated wetlands coalesce with the loss of permafrost (Connon et al., 2014). As permafrost degrades, raised plateaus subside resulting in the hydrological connection of previously isolated wetlands to the drainage network. This process significantly

alters the hydrological function of the different land cover types and the partitioning of water throughout the landscape (Quinton et al., 2011). Incremental additions of former isolated collapse scar bogs to the drainage network through the process of '*bog capture*' (Connon et al., 2014) result in transient contributions to basin runoff as the storage function of these wetlands is lost (Haynes et al., under review). Permanent contributions to runoff from transitioning discontinuous permafrost basins result from the expansion of the drainage area, and account for the greatest contribution to the observed

increases in runoff (Connon et al., 2014; Haynes et al., under review).

Field studies focusing on hydrology, meteorology and related fields have been conducted at Scotty Creek, NWT since the mid-1990s, with year-round monitoring since 1999. The direct effects of warming on the rate and pattern of permafrost thaw, the impacts of extreme events such as forest fires and anthropogenic influences such as the introduction of seismic lines on

permafrost distribution are monitored in the Scotty Creek basin. In this paper, we present a hydrological and micrometeorological (including subsurface temperature and soil moisture) dataset from the Scotty Creek basin for the period of 1 October, 2014 to 30 September, 2015 (*i.e.* 2015 water year), as part of the Special Observation and Analysis Period (SOAP) initiative coordinated by the Changing Cold Regions Network (CCRN). Data presented were collected from the



land cover types representative of the Scotty Creek basin including peat plateaus (both thawing and stable), bogs and channel fens.

## 2 Site Description

5    The Scotty Creek Research Station is located within the sporadic discontinuous permafrost zone in the NWT, Canada (61.44°N, 121.25°W), approximately 50 km south of Fort Simpson (Figure 1a, b). The mean annual air temperature (1981-2010) in Fort Simpson is -2.8°C with mean annual precipitation of 390 mm, of which approximately 38% (149 mm) falls as snow (Meteorological Service of Canada, 2017). Temperatures in this region have been steadily increasing, particularly during the winter months (Vincent et al., 2015), while annual precipitation has remained steady.

The Scotty Creek watershed covers 152-km$^2$ of subarctic boreal forest in the Taiga Plains ecozone. The watershed is comprised of heterogeneous upland moraines (48%), raised permafrost plateaus (20%), ombrotrophic bogs (19%), channel fens (12%), and lakes (2%) (Chasmer et al., 2014). The forested peat plateaus underlain by permafrost are dominated by an overstory of black spruce (*Picea mariana*), with an understory of Labrador tea (*Rhododendron groenlandicum*) and other

ericaceous shrubs, lichens (*Cladonia* spp.), and mosses (*Sphagnum* spp.). Channel fens are dominated by floating vegetative mats comprised of predominantly *Carex* and *Eriophorum* sedges with individual tamarack (*Larix laricina*) and birch (*Betula glandulosa*) trees scattered throughout the fens (Garon-Labrecque et al., 2015). Bogs in this basin are vegetated with ericaceous shrubs including leatherleaf (*Chamaedaphne calyculata*), bog rosemary (*Andromeda polifolia*), and small cranberry (*Vaccinium oxycoccos*) (Garon-Labrecque et al., 2015). The dominant bryophyte species in the bogs include

*Sphagnum balticum* and *S. magellanicum* (Garon-Labrecque et al., 2015). Peat deposits in the basin range from a depth of 2 to 8 m (McClymont et al., 2013).

Permafrost at the study site is present only beneath raised peat plateaus and is maintained as a result of insulation by overlying peat. These peat plateaus act as runoff generators as their capacity for water storage is low (Wright et al., 2009).

Runoff from plateaus is directed to adjacent wetlands as supra-permafrost subsurface flow where it may be stored or further conveyed to the drainage network. As permafrost beneath peat plateaus degrades, the ground surface subsides and becomes inundated, resulting in the transition of raised plateaus to waterlogged bogs and fens. Bogs isolated from the drainage network (*i.e.* channel fen networks) by remaining permafrost plateau barriers store water on the landscape, while bogs connected to the drainage network due to a breach in the surrounding permafrost contribute water to basin runoff (Connon et

al., 2014). Wide and hydraulically rough channel fens convey water through the watershed to the basin outlet (Hayashi et al., 2004).



## 3 Micrometeorological Station Data

Micrometeorological datasets are presented from four tripod stations located in land cover types representative of the Scotty Creek basin – bog, fen, stable peat plateau and thawing peat plateau (Figure 1c). The thawing peat plateau is undergoing rapid permafrost thaw and is actively transitioning to wetland. This dataset provides a contrast to the relatively unchanged
plateau on which the stable plateau tripod is located.

All data was recorded using Campbell Scientific dataloggers (CR10X) with a measurement interval of 60 seconds and averaged data output every 30 minutes. A continuous record over the SOAP year is presented for each station, with the exception of the fen tripod, which has gaps in the data due to programming and power issues. Each station was equipped to
measure net radiation, air temperature, and relative humidity. Four-component radiation (incoming and outgoing shortwave and longwave radiation) was measured with a net radiometer (Kipp & Zonen, CNR1 in bog and plateaus; NR-Lite in fen). Annual net radiation ($Q^*$) over the 2014-2015 water year was 1574 MJ m$^{-2}$ in the bog, 893 MJ m$^{-2}$ on the stable plateau, and 1528 MJ m$^{-2}$ on the thawing plateau, where the variation is a result of differences in canopy coverage (Figure 2). Air temperature and relative humidity were measured using a temperature and relative humidity probe (Campbell Scientific,
HMP45C) mounted near the top of the six-foot tripods and housed in a passively ventilated Gill radiation shield (Figure 2). Mean annual air temperature measured at the bog, stable plateau and thawing plateau was -1.9, -2.0 and -1.4°C, respectively. Snow depth was also measured on the bog, fen and thawing plateau using an SR50A snow depth sensor (Campbell Scientific) and this data has been corrected for temperature. Wind speed in the bog and fen (Figure 2) was measured by a Met One Instruments wind speed sensor (Model #014A). The height of all instruments located on the micrometeorological
tripods can be found in Table 1.

Additionally in the bog and thawing plateau ground temperature, soil moisture and ground heat flux were measured at several depths. To monitor ground temperatures in the bog, thermistors (Campbell Scientific, 107B) were installed at 10, 20, 30, 40, 50, 60, 80 and 100 cm below the ground surface (Figure 3a). In an excavated and back-filled soil pit located
centrally on the thawing plateau (see Hayashi et al., 2007 for details), ground temperatures were measured with thermistors installed at 0, 5, 10, 15, 20, 25, 30, 40, 50, 60 and 70 cm below the ground surface (Figure 3d). Soil moisture in the bog was measured at 10 and 20 cm below the ground surface (Campbell Scientific, HydraProbe II Soil Moisture Sensor) (Figure 3b). Using calibrated (Hayashi et al., 2007) water content reflectometers (WCR) (Campbell Scientific, CS615), volumetric water content at 10, 20, 30 40, and 50 cm below the ground surface was monitored in the thawing plateau (Figure 3e). In both the
bog and thawing plateau, ground heat flux ($Q_G$, W m$^{-2}$) was measured using a ground heat flux plate (Campbell Scientific, HFT3) installed at 5 cm below the ground surface (Figures 3c and 3f), calibrated according to the method described by Hayashi et al. (2007) and calculated to include heat storage between the ground surface and ground heat flux plate.



## 4 Deep Ground Temperatures

To monitor ground temperatures at depth, deep thermistor (RBR, XR-420) strings were installed in drilled boreholes and recorded temperature data hourly. Deep ground temperature datasets presented in this paper were located in a fen, a disturbed linear seismic line (cut in 1985), and in two locations beneath an old winter road – one site with standing water (Winter Road Wet) and one without (Winter Road Dry) (see Figure 1c for measurement locations, Figure 4 for data).

## 5 Precipitation

### 5.1 Total Precipitation

Total precipitation, both rain and snow, was measured at half-hourly intervals using an Alter-shielded Geonor precipitation gauge (Model T200B). Data presented have been corrected for wind undercatch according to the method of Smith (2007). Total precipitation depth was recorded at 30-minute intervals. Daily precipitation (rain, snow and total) is presented in Figure 5. In the 2014-2015 water year, 494 mm of total precipitation was received in the Scotty Creek basin, with approximately 45% of this (221 mm) occurring as snow.

### 5.2 Snow Surveys

Snowpack snow water equivalent (SWE) was determined prior to snowmelt in 2015 along established snow course transects, which traverse the representative land cover types including bogs, fens and peat plateaus. At intervals of 1 to 5 m, snow depth was measured using a steel ruler. Snow water equivalent was determined at every fifth depth measurement point using a Prairie-type snow sampling tube (Geoscientific) (inner diameter: 6.18 cm) and hanging scale.

## 6 Water Level

Water levels of individual wetlands and peat plateaus were monitored in single slotted stilling wells located at each site. Each well was equipped with either a Solinst (Levelogger Edge M2) or HOBO (U20-001-04) total pressure transducer. Data was recorded at 30-minute intervals and has been corrected for barometric pressure (Solinst Barologger Gold) in buffered thermal conditions (McLaughlin and Cohen, 2011). Water level data presented in this paper are from land cover types representative of the basin including a channel fen (Fen), a thawing peat plateau (Thawing Plateau), a bog isolated from the drainage network (Isolated Bog) and two bogs with differing degrees of drainage network connection (Fully Connected and Partially Connected Bog) (see Figure 1c). The pressure transducer in the channel fen was lowered in the well over the winter season to prevent sensor freezing and provide a continuous record for this water year (Figure 6). Pressure transducers were installed in the Partially Connected Bog and Thawing Plateau wells on 03 May 2015, while those in the Fully Connected and Isolated Bog were installed on 21 May 2015, once the wells were ice-free (Figure 6). Water level records for the 2015

growing season are provided in this dataset. Pressure transducer data was related to the manual measurements from the top of the well casing (ToC) to the water table at sensor installation and confirmed with the same measurements at sensor removal. The water level record standardized with the installation manual measurements was within $0.9 \pm 1.1$ cm (n = 5 sites in 2015) of the manual measurements at sensor removal. The water level data is presented in units of metres above sea

level (m asl) as well top of casing positions were surveyed in May 2015 using a differential global positioning system (SR530 RTK, Leica Geosystems Inc.).

## 7 Data Availability

All data presented in this paper are available from the Wilfrid Laurier University Library Research Data Repository (https://doi.org/10.5683/SP/OQDRJG).

## 8 Summary

The data presented in this paper comprise one year of the long-term hydrological, micrometeorological and geophysical research conducted in the Scotty Creek basin examining the effects of permafrost loss on ecosystem change. In coordination with SOAP year data from other cold regions sites within the Changing Cold Regions Network, this dataset may be used for comparative studies and modelling efforts to investigate the spatio-temporal variability in hydrological responses to

micrometeorological factors.

## 9 Acknowledgements

We acknowledge the contributions of all researchers involved in data collection at Scotty Creek. We also wish to thank the Dehcho First Nation for their support and acknowledge that the Scotty Creek Research Station is located on Treaty 11 land. We acknowledge funding support from the Changing Cold Regions Network (CCRN) and a Natural Sciences and

Engineering Research Council of Canada (NSERC) Discovery Grant.

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



**Table 1: Height of instruments located on the four micrometeorological tripods. Sensor heights and depths are expressed in m above or below ground surface, unless otherwise indicated. *For sensors below the ground, depth is given as a negative value.**

| Station | Sensor | Sensor height or depth* (m) |
| --- | --- | --- |
| **Bog Tripod** | | |
| | Four Component Radiometer (CNR1) | 1.45 |
| | Temperature and Relative Humidity Probe (HMP45C) | 1.90 |
| | Met One Wind Speed Sensor | 2.00 |
| | Snow Depth Sensor (SR50A) | 1.11 |
| | Ground Heat Flux Plate (HFT3) | - 0.05 m |
| | Thermistors (107B) | - 0.1 m |
| | | - 0.2 m |
| | | - 0.3 m |
| | | - 0.4 m |
| | | - 0.5 m |
| | | - 0.6 m |
| | | - 0.8 m |
| | | - 1.0 m |
| | Soil Moisture Sensor (HydraProbe II) | - 0.1 m |
| | | - 0.2 m |
| **Fen Tripod** | | |
| | Net Radiometer (NR-Lite) | 2.30 |
| | Temperature and Relative Humidity Probe (HMP45C) | 2.33 |
| | Met One Wind Speed Sensor | 2.53 |
| | Snow Depth Sensor (SR50A) | 1.00 |
| **Stable Plateau Tripod** | | |
| | Four Component Radiometer (CNR1) | 2.00 |
| | Temperature and Relative Humidity Probe (HMP45C) | 2.00 |
| **Thawing Plateau Tripod** | | |
| | Four Component Radiometer (CNR1) | 1.85 |



| | |
|---|---|
| Temperature and Relative Humidity Probe (HMP45C) | 2.15 |
| Snow Depth Sensor (SR50A) | 1.43 |
| Ground Heat Flux Plate (HFT3) | - 0.05 m |
| Thermistors (107B) | 0.0 m |
| | - 0.05 m |
| | - 0.1 m |
| | - 0.15 m |
| | - 0.2 m |
| | - 0.25 m |
| | - 0.3 m |
| | - 0.4 m |
| | - 0.5 m |
| | - 0.6 m |
| | - 0.7 m |
| Water Content Reflectometers (CS615) | - 0.1 m |
| | - 0.2 m |
| | - 0.3 m |
| | - 0.4 m |
| | - 0.5 m |



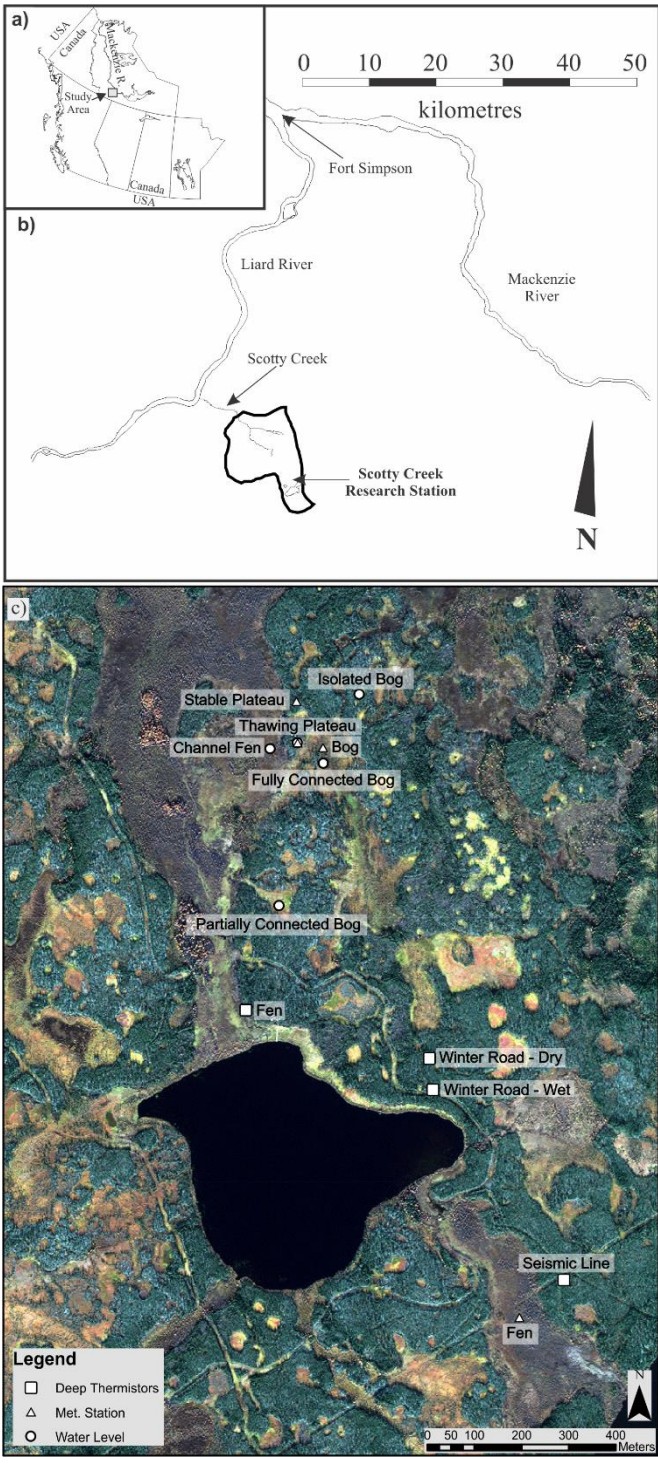

**Figure 1: (a) Location of study region in Canada, (b) map of Scotty Creek, NWT, and (c) a 2010 WorldView2 image of the 2.5 km²
area of study within Scotty Creek noting the locations of the instrumentation presented in the dataset.**







**Figure 2: (a) Mean daily air temperature (°C), (b) mean daily relative humidity (%), (c) total daily net radiation (Q\*, MJ m⁻²), and (d) mean daily wind speed (m s⁻¹) measured over the 2014-2015 water year at four micrometeorological tripods located on**
5  **representative land cover types in the Scotty Creek basin.**





**Figure 3:** Mean daily ground temperatures measured at the bog (a) and thawing plateau (d), mean daily soil moisture (expressed as volumetric water content, VWC) for the bog ((b) with corresponding ground temperature) and thawing plateau (e), and total daily ground heat flux (W m$^{-2}$) for the bog (c) and thawing plateau (f).

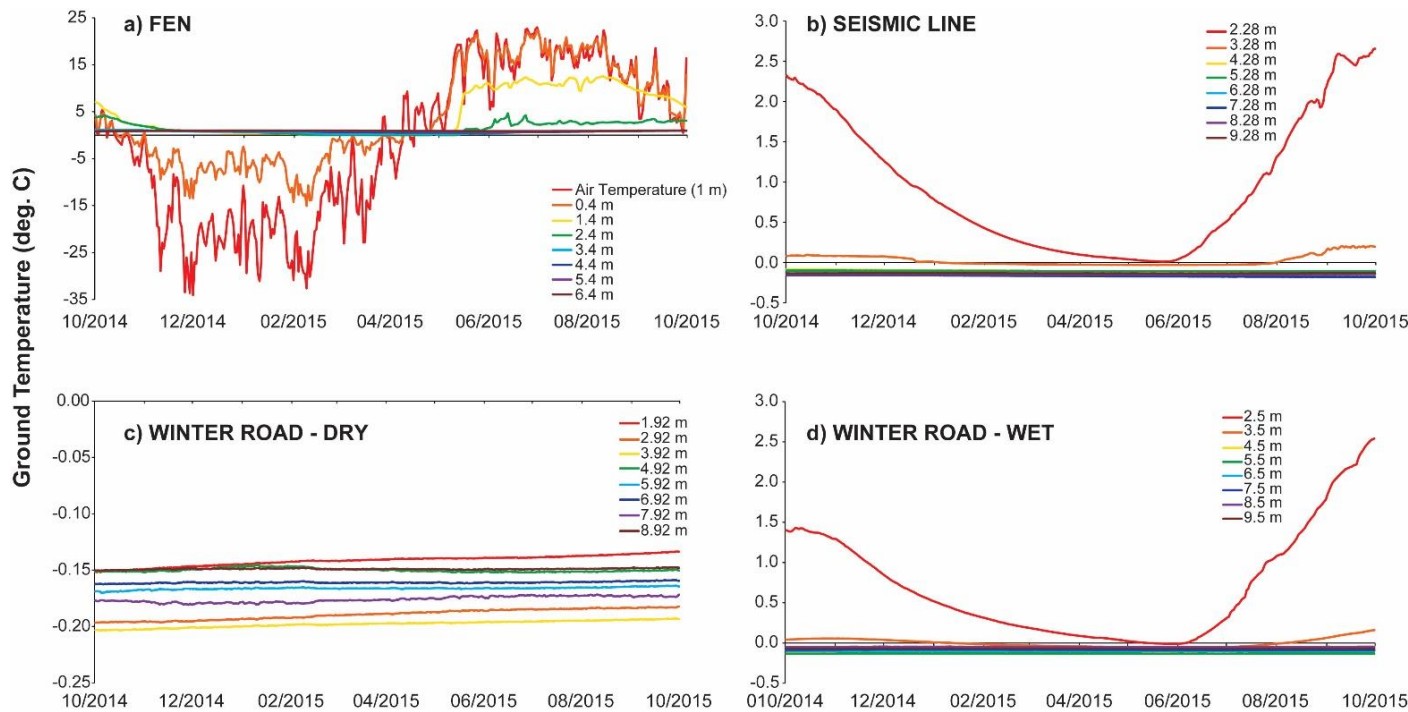

**Figure 4:** Deep mean daily ground temperatures measured with thermistor strings located (a) in the fen, (b) beneath a seismic line, (c) beneath a winter road with no standing water present and (d) beneath a winter road with standing water present. Depth of measurements are metres below ground surface, except air temperature measured 1 m above the ground surface in the fen. Note difference in temperature scale for each site.




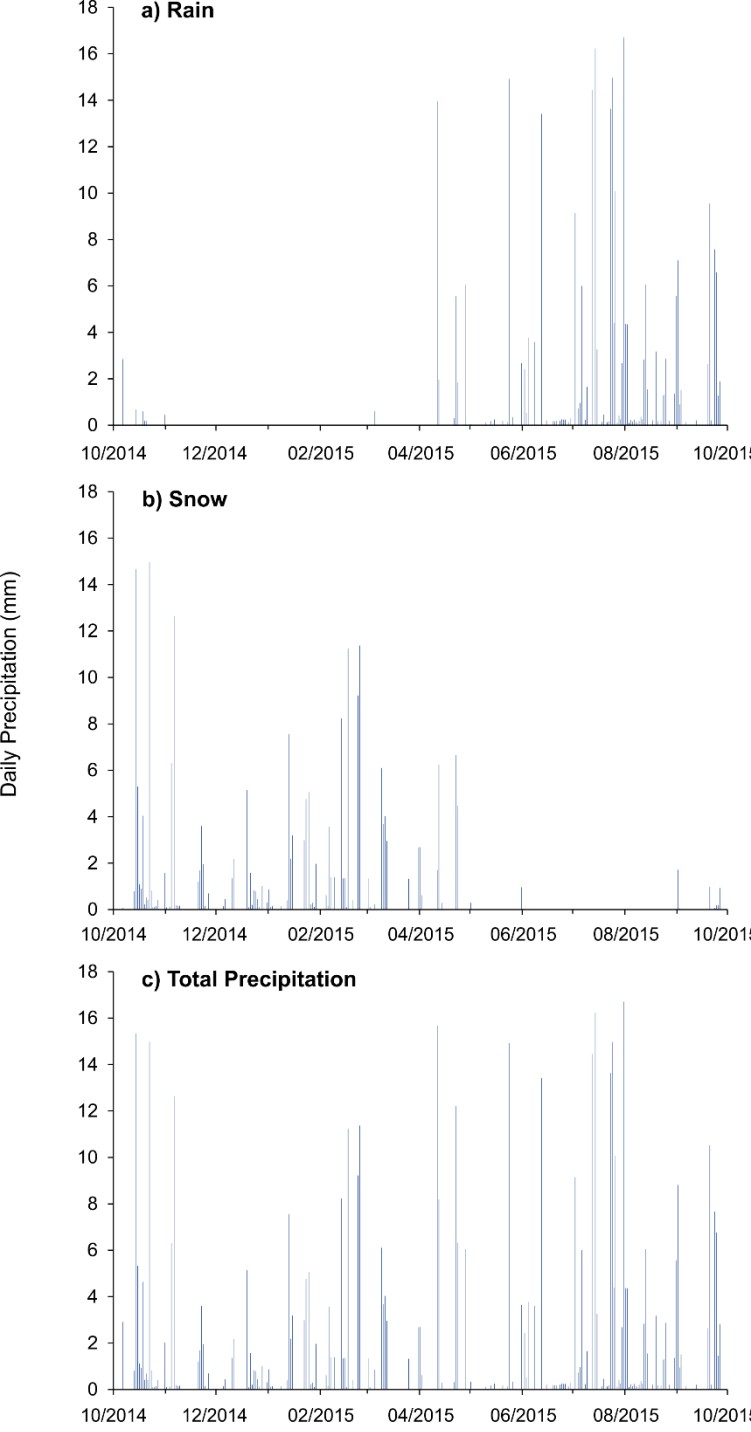

**Figure 5:** Daily (a) rain, (b) snow, and (c) total precipitation (all in mm) measured with the Geonor precipitation gauge during the 2014-2015 water year.



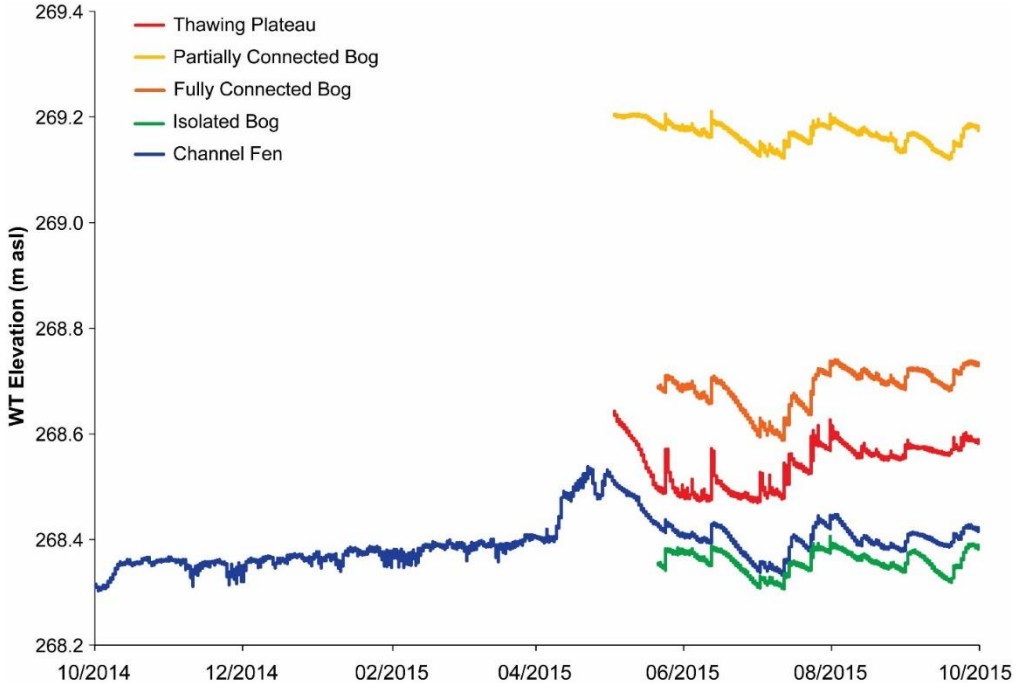

**Figure 6: Water table elevation expressed in metres above sea level (m asl) in five locations representative of the land cover types comprising the Scotty Creek basin.**