# Peer review of "Hydrometric measurements in peatland-dominated, discontinuous permafrost at Scotty Creek, Northwest Territories, Canada -Changing Cold Regions Network (CCRN) Special Observation and Analysis Period (SOAP)"

_Earth System Science Data, 2018_

## Referee Comment (RC1) · Anonymous Referee #1 · 12 Jul 2018

Review of "Hydrometric measurements in peatland-dominated, discontinuous permafrost at Scotty Creek, Northwest Territories, Canada - Changing Cold Regions Network (CCRN) Special Observation and Analysis Period (SOAP)" by Kristine M. Haynes, Ryan F. Connon, and William L. Quinton. The presented dataset covers soil temperatures, soil moisture and groundheat fluxes as well as meteorological variables for

multiple landscape elements within the Scotty Creek catchment, Canada. The data covers about 1 year and looks interesting from the presented figures. Unfortunately, I couldn't judge the data itself as it is under embargo until 2019 and I did not have the time nor the intention to disclose my anonymity to go and ask for the data as suggested in the data-repository. Therefore I can't really review this dataset except for the presented figures. Overall, the manuscript is well written and the relevance of the dataset is clear from the introduction. However, why exactly the authors chose to present this data (the year 2015) from an ongoing measurement effort since 1990, and if more data is or becomes available, does not become clear from the manuscript.

The data seems of high quality although they also raise questions: - What were the research goals for collecting the presented data in the first place? This does not become clear from the introduction. - Data quality and data gaps are not addressed in the manuscript (%of time coverage, number of gaps etc), except for page 4, line 8,9 that states that some gaps exist. - Why doesn't the soil temperature in the bog go below zero, while ice lenses forming in bogs are quite common. Is this data correct?

Minor comment: Pag1 line 15 Micro. . .. Presented. Multiple interpretations possible. Please rephrase. Page 2 line 21 collapse –> collapsed. Explain a bit more here.

---

## Referee Comment (RC2) · Anonymous Referee #2 · 15 Jul 2018

Review of "Hydrometric measurements in peatland-dominated, discontinuous permafrost at Scotty Creek, Northwest Territories, Canada – Changing Cold Regions Network (CCRN) Special Observation and Analysis Period (SOAP)" by Kristine M. Haynes, Ryan F. Connon, and William L. Quinton

Submitted to Earth System Science Data - Manuscript Number: ESSD-2018-68

[Figure]

Summary: This brief communication is an interesting and highly suitable contribution to Earth System Science Data (ESSD). The paper is generally well-written and figures are very clear and entirely appropriate to illustrate key aspects of the dataset. This report provides guidance that the authors should consider in revising their manuscript.

General Comments:

1) One requirement for publication in ESSD is the inclusion of clear statements on the limitations of the datasets, which are lacking in the paper. For instance, what are the operating range, accuracy and precision of the instrumentation used? Are there gaps in the datasets and was in-filling performed on these gaps (if any)? Was there any quality control/analysis performed on the data? In any case, information on the limitations of the observational data should be included in a revised paper.

2) Similarly, the journal requires the datasets to be openly accessible to warrant publication. While the datasets reported in this paper are indeed available in an online data repository at Wilfrid Laurier University, there is an embargo on the data until 31 May 2019. Is it therefore too soon to publish this article when the data are not readily accessible? This is particularly a concern if one wishes to verify the quality of the datasets, which should be deemed sufficient to warrant publication.

3) The journal requires statements on the author contributions and competing interests, as well as a special issue statement, prior to the Acknowledgements.

Specific Comments:

1) P. 1, lines 1-4: The title should better reflect the datasets described in this paper. Perhaps the title should thus be modified to "Hydrometeorological measurements . . ." or "Micrometeorological measurements . . ." Further to this, is the second part of the title necessary? If so, then the en dash should be replaced with "during the".

2) P. 1, line 15: Change to "1 October".

3) P. 1, line 26: Replace "dramatic" with "substantial" or another similar word.
4) p. 2, line 16: Use "terrain" in the singular form.

5) p. 2, line 32: Delete the commas after the months.

6) P. 4, line 7: Change to "All data were".

7) P. 4, line 18: Revise to "these data have"

8) p. 5, lines 1-5: At what depth are the 'deep groundwater temperatures' being measured? How to they compare to the depths of the soil temperatures reported in Section 3?

9) P. 5, lines 11-12: How was the partitioning of the precipitation into its liquid and solid component achieved? Is this based solely on a $0°C$ threshold?

10) P. 5, line 13: Was there only one snow survey conducted or multiple ones prior to snowmelt? On what dates were these snow surveys conducted? What were the average snow depths/snow water equivalents during the snow surveys? What is the precision of these snow measurements and possible sources of errors? More information on this component of the hydrometeorological datasets is needed.

11) P. 5, line 21: Revise the text to: "were recorded at 30-minute intervals and were corrected . . ."

12) P. 5, line 27: Change to "3 May".

13) P. 6, line 1: Change to "were related".

14) P. 6, line 4: Change to "are presented".

15) P. 6, line 8: In what format are the datasets archived? What metadata are available with these files?

16) P. 7, line 15: Update this reference with an article number or page range if possible.

17) P. 7, line 31: Update the status of this reference if possible.

18) P. 9, line 1: The journal may require all authors on this reference be listed here instead of "et al."

19) P. 9, line 6: Add the paper number for this article.

20) P. 10, Table 1: Consider adding the operating range, precision and accuracy of each instrument to the table.

21) P. 14, Figure 3: How well do the water content reflectometers operate in frozen ground? What may be the source of the spike in ground heat flux at the bog in spring 2015, or are these spurious measurements?

---

## Author Comment (AC1) · 5 Sep 2018

**Hydrometric measurements in peatland-dominated, discontinuous permafrost at Scotty Creek, Northwest Territories, Canada - Changing Cold Regions Network (CCRN) Special Observation and Analysis Period (SOAP)**

Kristine M. Haynes, Ryan F. Connon and William L. Quinton

Cold Regions Research Centre, Wilfrid Laurier University, Waterloo, N2L 3C5, Canada

*Correspondence to*: Kristine M. Haynes (khaynes@wlu.ca)

**Response to Anonymous Referee #1 Comments**

**Authors' Comments are in bold and denoted with 'AC'.**

The presented dataset covers soil temperatures, soil moisture and ground heat fluxes as well as meteorological variables for multiple landscape elements within the Scotty Creek catchment, Canada. The data covers about 1 year and looks interesting from the presented figures. Unfortunately, I couldn't judge the data itself as it is under embargo until 2019 and I did not have the time nor the intention to disclose my anonymity to go and ask for the data as suggested in the data-repository. Therefore I can't really review this dataset except for the presented figures. Overall, the manuscript is well written and the relevance of the dataset is clear from the introduction. However, why exactly the authors chose to present this data (the year 2015) from an ongoing measurement effort since 1990, and if more data is or becomes available, does not become clear from the manuscript.

**Authors' Comment (AC): We thank Anonymous Referee #1 for their positive comments.**

**Access to the data is granted following registration for log-in information including during the embargo period. As we understand, this aligns with the Earth System Science Data journal Repository Criteria, which specifies that "a usual registration to get a login free-of-charge" may be in place to gain access to the data. This simple registration was put in place to monitor use of our data and facilitate the potential for collaborative research.**

**We have selected to present the data from only the 2015 water year (1 October 2014 to 30 September 2015) as this year represents the Special Observation and Analysis Period (SOAP) established by the Changing Cold Regions Network (CCRN), as we mention throughout the paper. A main goal set forth among the participants of the CCRN at the conclusion of the initiative (completed March 2018) was the publication and synthesis of hydrological and meteorological datasets from study sites across the network over a**

common time frame. Facilitating this coordination of CCRN SOAP datasets is the Earth System Science Data "Water, ecosystem, cryosphere, and climate data from the interior of Western Canada and other cold regions" Special Issue, to which this Brief Communication was submitted. The overarching objective of the synthesis effort, for which all CCRN datasets will be available, is to examine the spatial and temporal variability in local hydrological responses to meteorological influences across cold regions. We aim to clarify our justification for the selection of this year of data in the revised manuscript on Page 3 Lines 4-5.

The data seems of high quality although they also raise questions: - What were the research goals for collecting the presented data in the first place? This does not become clear from the introduction. - Data quality and data gaps are not addressed in the manuscript (%of time coverage, number of gaps etc), except for page 4, line 8,9 that states that some gaps exist. - Why doesn't the soil temperature in the bog go below zero, while ice lenses forming in bogs are quite common. Is this data correct?

AC: As mentioned in the Introduction of the manuscript, field research has been conducted in the Scotty Creek basin since the mid-1990s, with year-round monitoring since 1999. The aim of this long-term and ongoing research is to explore the dominant hydrological processes occurring in the discontinuous permafrost landscape and examine how permafrost thaw is changing the routing and storage of moisture in this landscape. Research has also been, and continues to be, undertaken to investigate the impact of natural and anthropogenic disturbances on hydrological cycling in this region, including the impacts of forest fires and the cutting of linear disturbances (*i.e.* seismic lines and winter roads) for transportation and resource exploration. Year-round hydrological and meteorological monitoring is conducted with these objectives in mind. The dataset presented in this manuscript represents one year of annual measurements from the representative land cover types of the Scotty Creek headwaters (bogs, fens, peat plateaus) and includes data from instrumentation monitoring linear disturbances (seismic lines and former winter roads). This is reinforced in the revised manuscript.

In terms of data gaps, the only gaps in data from the micrometeorological stations appear in the channel fen record, due to the programming and power issues mentioned in the manuscript. The period of available data from the channel fen micrometeorological station is evident in the plots of Figure 2. Given the length of the channel fen station gap, no attempt was made to gap-fill. Additionally, the bog soil moisture and temperature record at 10 cm below the ground surface ends approximately 2 months prior to the end of the annual record due to a sensor error. This data gap is mentioned in the revised manuscript. No other data gaps occurred during the presented dataset.

The ground temperatures at 10 cm below the surface in the bog reach a minimum of -0.2°C during the winter months. There are no errors in this dataset. This trend is reinforced by a similar trend in ground temperature measured in association with the soil moisture measurements and are presented in Figures 3a and 3b. The soil moisture data presented in Figure 3b suggests that the bog does indeed freeze at both 10 and 20 cm. The temperature at these depths remains relatively isothermal over winter as additional energy loss from the bog is consumed as latent heat to freeze the water underlying the freezing front.

**Measurements from drilling observations in late winter confirm a typical re-freeze depth of 20-30 cm in the bog.**

5    Minor comment: Pag1 line 15 Micro…. Presented. Multiple interpretations possible.

**AC: This statement has been clarified.**

Please rephrase. Page 2 line 21 collapse –> collapsed. Explain a bit more here.

**AC: The word 'collapse' in this sentence is associated with the term 'collapse scar bogs' and therefore**
10    **should not be 'collapsed'. Collapse scar bogs are common thermokarst features in permafrost peatland environments. These features are located within raised peat plateaus as the underlying permafrost degrades. Collapse scar bogs typically function as depressional storage features on the landscape, as raised permafrost surrounding the bogs prevents flow. The definition of a collapse scar bog has been incorporated into the sentence on Page 2 Lines 18-19.**

15

---

## Author Comment (AC2) · 5 Sep 2018

**Hydrometric measurements in peatland-dominated, discontinuous permafrost at Scotty Creek, Northwest Territories, Canada - Changing Cold Regions Network (CCRN) Special Observation and Analysis Period (SOAP)**

Kristine M. Haynes, Ryan F. Connon and William L. Quinton

Cold Regions Research Centre, Wilfrid Laurier University, Waterloo, N2L 3C5, Canada

*Correspondence to*: Kristine M. Haynes (khaynes@wlu.ca)

**Response to Anonymous Referee #2 Comments**

**Authors' Comments are in bold and denoted with 'AC'.**

Summary: This brief communication is an interesting and highly suitable contribution to Earth System Science Data (ESSD). The paper is generally well-written and figures are very clear and entirely appropriate to illustrate key aspects of the dataset. This report provides guidance that the authors should consider in revising their manuscript.

**Authors' Comment (AC): We thank Anonymous Referee #2 for their positive comments. Individual comments are addressed below and changes are tracked in the revised manuscript.**

General Comments:

1) One requirement for publication in ESSD is the inclusion of clear statements on the limitations of the datasets, which are lacking in the paper. For instance, what are the operating range, accuracy and precision of the instrumentation used? Are there gaps in the datasets and was in-filling performed on these gaps (if any)? Was there any quality control/analysis performed on the data? In any case, information on the limitations of the observational data should be included in a revised paper.

**AC: Gaps in the datasets have been detailed with the description of the data in the revised manuscript. Given the length of the data gaps, no attempt has been made to in-fill these gaps – this has now been specified in the manuscript. Greater detail pertaining to instrumentation precision and accuracy has now been incorporated into Table 1. The data were checked for quality control and any calculations or**

**corrections applied to the data are mentioned with the description of the data collection. A statement clarifying this is now on Page 3 Lines 9-10.**

2) Similarly, the journal requires the datasets to be openly accessible to warrant publication. While the datasets reported in this paper are indeed available in an online data repository at Wilfrid Laurier University, there is an embargo on the data until 31 May 2019. Is it therefore too soon to publish this article when the data are not readily accessible? This is particularly a concern if one wishes to verify the quality of the datasets, which should be deemed sufficient to warrant publication.

**AC: The datasets are currently available to be accessed through the Wilfrid Laurier University Library Research Data Repository. Access to the data is granted following registration for log-in information. As we understand, this aligns with the Earth System Science Data journal Repository Criteria, which specifies that "a usual registration to get a login free-of-charge" may be in place to gain access to the data. Our datasets may be accessed via this process at any time including during the specified embargo period. This simple registration was put in place to monitor use of our data and facilitate the potential for collaborative research.**

3) The journal requires statements on the author contributions and competing interests, as well as a special issue statement, prior to the Acknowledgements.

**AC: Statements on Author Contributions and Competing Interests have now been included prior to the Acknowledgements. We understand from the Manuscript Preparation Guidelines that a Special Issue Statement will be included by Copernicus.**

Specific Comments:

1) P. 1, lines 1-4: The title should better reflect the datasets described in this paper. Perhaps the title should thus be modified to "Hydrometeorological measurements…" or "Micrometeorological measurements…" Further to this, is the second part of the title necessary? If so, then the en dash should be replaced with "during the".

**AC: As suggested, the title has been changed to "Hydrometeorological measurements in peatland-dominated, discontinuous permafrost at Scotty Creek, Northwest Territories, Canada during the Changing Cold Regions Network (CCRN) Special Observation and Analysis Period (SOAP)".**

2) P. 1, line 15: Change to "1 October".

**AC: Changed.**

3) P. 1, line 26: Replace "dramatic" with "substantial" or another similar word.

**AC: Replaced with "substantial" as suggested.**

4) p. 2, line 16: Use "terrain" in the singular form.

**AC: Corrected.**

5) p. 2, line 32: Delete the commas after the months.

**AC: Deleted.**

6) P. 4, line 7: Change to "All data were".

10 **AC: Corrected.**

7) P. 4, line 18: Revise to "these data have"

**AC: Corrected.**

15 8) p. 5, lines 1-5: At what depth are the 'deep groundwater temperatures' being measured? How to they compare to the depths of the soil temperatures reported in Section 3?

**AC: The depths at which deep ground temperatures were measured at these disturbed sites are now specified in this section (Page 5 Lines 6-10).**

20 9) P. 5, lines 11-12: How was the partitioning of the precipitation into its liquid and solid component achieved? Is this based solely on a 0°C threshold?

**AC: Precipitation type was determined based on hydrometeor temperature as described by Harder and Pomeroy (2014). This information has now been included in the explanation of wind undercatch calculations on Lines 14-16 on Page 5 and this reference has been included in the reference list.**

25

10) P. 5, line 13: Was there only one snow survey conducted or multiple ones prior to snowmelt? On what dates were these snow surveys conducted? What were the average snow depths/snow water equivalents during the snow surveys? What is the precision of these snow measurements and possible sources of errors? More information on this component of the hydrometeorological datasets is needed.

30 **AC: Snow survey data presented here was collected 23-24 March 2015 just prior to snowmelt. Greater detail of the snow depth and snow water equivalents measured across the fen, bog and peat plateau land covers has now been provided in this section on Pages 5-6.**

11) P. 5, line 21: Revise the text to: "were recorded at 30-minute intervals and were corrected…"

**AC: Changed as suggested.**

12) P. 5, line 27: Change to "3 May".

5 **AC: Changed.**

13) P. 6, line 1: Change to "were related".

**AC: Changed.**

10 14) P. 6, line 4: Change to "are presented".

**AC: Changed.**

15) P. 6, line 8: In what format are the datasets archived? What metadata are available with these files?

**AC: The archived datasets are tab-delimited files, which can be dealt with using Microsoft Excel. Each of**
15 **the four datasets contains a metadata worksheet defining the data columns, applicable units and other**
**pertinent information. Greater detail on the dataset files is now provided in the Data Availability section.**

16) P. 7, line 15: Update this reference with an article number or page range if possible.

**AC: The page range has now been added to this reference.**

20

17) P. 7, line 31: Update the status of this reference if possible.

**AC: The status of this paper has not changed.**

18) P. 9, line 1: The journal may require all authors on this reference be listed here instead of "et al."

25 **AC: The full author list has now been included in this reference.**

19) P. 9, line 6: Add the paper number for this article.

**AC: The paper number has been added and the page range removed for this reference.**

30 20) P. 10, Table 1: Consider adding the operating range, precision and accuracy of each instrument to the table.

**AC: This information has been incorporated into Table 1.**

21) P. 14, Figure 3: How well do the water content reflectometers operate in frozen ground? What may be the source of the spike in ground heat flux at the bog in spring 2015, or are these spurious measurements?

**AC:  A sharp decrease in liquid volumetric soil moisture content indicates the onset of soil freezing. Liquid water soil content of frozen soils typically ranges from 0.15 to 0.2 (Connon et al., 2018).  This has**
5 **now been specified on Page 4 Line 33 to Page 5 Line 1.**

**The ground heat flux data was checked for quality control.  The large spike in the bog ground heat flux in spring 2015 is attributed to the loss of the insulating snowpack during the spring melt period.  The spring peak in ground heat flux coincides with the observed substantial increase in volumetric liquid soil moisture**
10 **back to saturation following ground thaw and infiltration of snowpack meltwater. The large spike in bog ground heat flux is also observed in all other years of available data during the spring.**

---

## Author Comment (AC3) · 5 Sep 2018

**Hydrometeorological<del>ric</del> measurements in peatland-dominated, discontinuous permafrost at Scotty Creek, Northwest Territories, Canada during the- Changing Cold Regions Network (CCRN) Special Observation and Analysis Period (SOAP)**

[revised manuscript text omitted]

Field studies focusing on hydrology, meteorology and related fields have been conducted at Scotty Creek, NWT since the mid-1990s, with year-round monitoring since 1999. The aim of this long-term and ongoing research is to explore the dominant hydrological processes occurring in the discontinuous permafrost peatland landscape and examine the impacts of permafrost

30 thaw-induced land cover change on these hydrological mechanisms. The direct effects of warming on the rate and pattern of permafrost thaw, the impacts of extreme events such as forest fires and anthropogenic influences such as the introduction of seismic lines on permafrost distribution are monitored in the Scotty Creek basin. In this paper, we present a hydrological and micrometeorological (including subsurface temperature and soil moisture) dataset from the Scotty Creek basin for the period of 1 October7 2014 to 30 September7 2015 (*i.e.* 2015 water year), as part of the Special Observation and Analysis Period

(SOAP) initiative coordinated by the Changing Cold Regions Network (CCRN). Data presented were collected from the land cover types representative of the Scotty Creek headwatersbasin including peat plateaus (both thawing and stable), bogs and channel fens and includes data from instrumentation monitoring linear disturbances (seismic lines and former winter roads). In coordination with other SOAP year datasets across the CCRN, the overarching objective of this data contribution is to

5 examine the spatial and temporal variability in local hydrological responses to meteorological influences across cold regions.

**2 Site Description**

[revised manuscript text omitted]

**5 4 Deep Ground Temperatures**

To monitor ground temperatures at depth, deep thermistor (RBR, XR-420) strings were installed in drilled boreholes and recorded hourly temperature data at approximately1 m depth intervals-hourly. Deep ground temperature datasets presented in this paper were located in a fento a maximum depth of 6.4 m, a disturbed linear seismic line (cut in 1985) down to 9.28 m, and in two locations beneath an old winter road – one site with standing water (Winter Road Wet) to a maximum depth of 9.5 m and one without (Winter Road Dry) down to 8.92 m (see Figure 1c for measurement locations, Figure 4 for data).

**5** Precipitation**

10

**5.1 Total Precipitation**

Total precipitation, both rain and snow, was measured at half-hourly intervals using an Alter-shielded Geonor precipitation gauge (Model T200B). Data presented have been corrected for wind undercatch according by determining precipitation type

15 from hydrometeor temperature and adjusting for the catch efficiency of the gauge depending on wind speed measured at the height of the altar shield (Harder and Pomeroy, 2014; Smith, 2007)-to the method of Smith (2007). Total precipitation depth was recorded at 30-minute intervals. Daily precipitation (rain, snow and total) is presented in Figure 5. In the 2014-2015 water year, 494 mm of total precipitation was received in the Scotty Creek basin, with approximately 45% of this (221 mm) occurring as snow.

**20 5.2 Snow Surveys**

Snowpack snow water equivalent (SWE) was determined prior to snowmelt in 2015 along established snow course transects, which traverse the representative land cover types including bogs, fens and peat plateaus. In an effort to effectively capture the spatial variability of snow distributionAt intervals of 1 to 5 m, snow depth was measured using a steel ruler at intervals of 1 to 5 m. Snow water equivalent was determined at every fifth depth measurement point using a Prairie-type snow sampling

25 tube (Geoscientific) (inner diameter: 6.18 cm) and hanging scale. Snow survey data presented here were collected on 23-24 March 2015, capturing snow depth and SWE just prior to the initiation of the snowmelt. Mean snow depth was 64.2 ± 10.8 (standard deviation) cm on fen land cover (n = 14), 69.5 ± 8.6 cm on bogs (n = 127), and 77.7 ± 10.8 cm on plateaus (n = 79). Snow density was similar across the land cover types with 0.204 ± 0.033 g cm-3 measured on fens (n = 5), 0.246 ± 0.036 g cm-3

 $\frac{3}{3}$  on bogs (n = 31) and 0.232 ± 0.038 g cm-3 on peat plateaus (n = 19). Therefore, SWE ranged from 13.1 cm in the fen snowpack up to 18.0 cm on peat plateaus, with bog SWE measured to be 17.1 cm.

**6** Water Level**

Water levels of individual wetlands and peat plateaus were monitored in single slotted stilling wells located at each site. Each
well was equipped with either a Solinst (Levelogger Edge M2) or HOBO (U20-001-04) total pressure transducer. Data wereas recorded at 30-minute intervals and has beenwere corrected for barometric pressure (Solinst Barologger Gold) in buffered thermal conditions (McLaughlin and Cohen, 2011). Water level data presented in this paper are from land cover types representative of the basin including a channel fen (Fen), a thawing peat plateau (Thawing Plateau), a bog isolated from the drainage network (Isolated Bog) and two bogs with differing degrees of drainage network connection (Fully Connected and Partially Connected Bog) (see Figure 1c). The pressure transducer in the channel fen was lowered in the well over the winter

- season to prevent sensor freezing and provide a continuous record for this water year (Figure 6). Pressure transducers were installed in the Partially Connected Bog and Thawing Plateau wells on  $\theta$ 3 May 2015, while those in the Fully Connected and Isolated Bog were installed on 21 May 2015, once the wells were ice-free (Figure 6). Water level records for the 2015 growing season are provided in this dataset. Pressure transducer data wereas related to the manual measurements from the top of the
- 15 well casing (ToC) to the water table at sensor installation and confirmed with the same measurements at sensor removal. The water level record standardized with the installation manual measurements was within  $0.9 \pm 1.1$  cm (n = 5 sites in 2015) of the manual measurements at sensor removal. The water level data areis presented in units of metres above sea level (m asl) as well top of casing positions were surveyed in May 2015 using a differential global positioning system (SR530 RTK, Leica Geosystems Inc.).

**20 7 Data Availability**

All data presented in this paper are available from the Wilfrid Laurier University Library Research Data Repository (https://doi.org/10.5683/SP/OQDRJG). The datasets are provided in four tab-delimited files. Each of the four dataset files contains a metadata worksheet defining the data columns, applicable units and other pertinent information.

**8 Summary**

25 The data presented in this paper comprise one year of the long-term hydrological, micrometeorological and geophysical research conducted in the Scotty Creek basin examining the effects of permafrost loss on ecosystem change. In coordination with SOAP year data from other cold regions sites within the Changing Cold Regions Network, this dataset may be used for

comparative studies and modelling efforts to investigate the spatio-temporal variability in hydrological responses to micrometeorological factors.

**Author Contributions**

5 KMH and RFC compiled the data, which was collected by RFC and WLQ. KMH prepared the manuscript with input from RFC and WLQ.

**Competing Interests**

The authors declare that they have no conflict of interest.

**10 9-Acknowledgements**

We acknowledge the contributions of all researchers involved in data collection at Scotty Creek. We also wish to thank the Dehcho First Nation for their support and acknowledge that the Scotty Creek Research Station is located on Treaty 11 land. We acknowledge funding support from the Changing Cold Regions Network (CCRN) and a Natural Sciences and Engineering Research Council of Canada (NSERC) Discovery Grant. We thank the two anonymous reviewers for their constructive comments.

**References**

[revised manuscript text omitted]

| Station       | Sensor                                              | Sensor height or
depth* (m)                                                       | Operating Range                         | Accuracy                                                                                                                                                                                                                                                                                                                                                    |
|---------------|-----------------------------------------------------|--------------------------------------------------------------------------------------|------------------------------------------------|-------------------------------------------------------------------------------------------------------------------------------------------------------------------------------------------------------------------------------------------------------------------------------------------------------------------------------------------------------------|
| Bog
Tripod |                                                     |                                                                                      |                                                |                                                                                                                                                                                                                                                                                                                                                             |
|               | Four Component
Radiometer (CNR1)                 | 1.45                                                                                 | 0.3-50 µт                               | ± 10%                                                                                                                                                                                                                                                                                                                                                |
|               | Temperature and Relative
Humidity Probe (HMP45C) | 1.90                                                                                 | Temp: -40 to +60°C
RH: 0-100% | $\frac{\text{Temp: } \pm 0.2^{\circ}\text{C @ } 20^{\circ}\text{C}}{\text{RH: } \pm 2\% @ 0 \text{ to } 90\% \text{ humidity}}$ $\pm 3\% @ 90 \text{ to } 100\% \text{ humidity}$                                                                                                                                                                           |
|               | Met One Wind Speed Sensor                           | 2.00                                                                                 | 0-60 m s-1                   | $\pm 0.11 \text{ m s}^{-1} \text{ or } 1.5\% \text{ FS}$                                                                                                                                                                                                                                                                                                    |
|               | Snow Depth Sensor
(SR50A)                        | 1.11                                                                                 | $-45 \text{ to } +50^{\circ}\text{C}$          | $\frac{\text{Maximum of} \pm 1 \text{ cm or } 0.4\% \text{ of distance}}{\text{to target}}$                                                                                                                                                                                                                                                                 |
|               | Ground Heat Flux Plate
(HFT3)                    | - 0.05 m                                                                             | $-40 \text{ to } +55^{\circ}\text{C}$          | better than $\pm$ 5% of reading                                                                                                                                                                                                                                                                                                                             |
|               | Thermistors (107B)                                  | - 0.1 m
- 0.2 m
- 0.3 m
- 0.4 m
- 0.5 m
- 0.6 m
- 0.8 m
- 1.0 m | -35 to +50°C                            | Worst case: $\pm 0.4^{\circ}C$ (-24 to 48°C) $\pm 0.9^{\circ}C$ (-35 to 50°C)Interchangeability Error: $\pm 0.10^{\circ}C$ (0 to 50°C) $\pm 0.20^{\circ}C$ at -10°C $\pm 0.20^{\circ}C$ at -10°C $\pm 0.30^{\circ}C$ at -20°C $\pm 0.40^{\circ}C$ at -30°C $\pm 0.50^{\circ}C$ at -40°CSteinhart-HartEquation Error: $\leq \pm 0.01^{\circ}C$ (-35 to 50°C) |
|               | Soil Moisture Sensor (Hydra
Probe II)            | - 0.1 m
- 0.2 m                                                                   |                                                |                                                                                                                                                                                                                                                                                                                                                             |

|         | Real dielectric permittivity |      | 1 to 80 where 1 = <math>\frac{1}{1}</math> to 80 where 1 = <math>\frac{1}{1}</math> = <math>\frac{1}{10}</math> | $\pm \le 1.5\%$ or 0.2 whichever is typically                                                    |
|---------|------------------------------|------|------------------------------------------------------------------------------------------------------------------------|--------------------------------------------------------------------------------------------------|
|         | (isolated)                   |      | air, 80 = distilled                                                                                                    | greater                                                                                          |
|         |                              |      | water                                                                                                                  |                                                                                                  |
|         |                              |      |                                                                                                                        |                                                                                                  |
|         | Soil moisture for inorganic  |      | From completely                                                                                                        | $\pm 0.01$ WFV for most soils                                                                    |
|         | & mineral soil    |      | dry to fully                                                                                                           | $\pm \le 0.03$ max for fine textured soils                                                       |
|         |                              |      | saturated                                                                                                              |                                                                                                  |
|         | Bulk electrical conductivity |      | 0.01 to 1.5 S m -1                                                                                          | Max. of <math>\pm 2.0\%</math> or 0.02 S m-1                                   |
|         |                              |      |                                                                                                                        |                                                                                                  |
|         | Temperature                  |      | $-10^{\circ}$ to $+55^{\circ}$ C                                                                                       | $\pm 0.3^{\circ} \text{ C}$                                                                      |
|         | Inter-sensor variability     |      | N/A                                                                                                             | $\pm$ 0.012 WFV ( $\theta$ m 3 m -3 )                                      |
| Fen     |                              |      |                                                                                                                        |                                                                                                  |
| Tripod  |                              |      |                                                                                                                        |                                                                                                  |
|         |                              |      | Spectral 0.2 to 100                                                                                                    | Directional error:                                                                               |
|         | Net Radiometer (NR-Lite)     | 2.30 | $\frac{\mu m (Measurement)}{\pm 2000 \text{ W m}^{-2}}$                                                                | $(0 - 60^{\circ} \text{ at } 1000 \text{ W m}^{-2}): < 30 \text{ W m}^{-2}$                      |
|         |                              |      |                                                                                                                        | Sensor asymmetry: $\pm 5\%$ typical, ( $\pm 10\%$                                                |
|         |                              |      |                                                                                                                        | worst case)                                                                                      |
|         | Temperature and Relative     |      | Temp: <math>-40</math> to <math>+60^{\circ}C</math>                                                             | $\frac{\text{Temp: } \pm 0.2^{\circ}\text{C} @ 20^{\circ}\text{C}}{20^{\circ}\text{C}}$          |
|         | Humidity Probe (HMP45C)      | 2.33 | RH: 0-100%                                                                                                      | $\underline{RH: \pm 2\% @ 0 to 90\% humidity}$                                                   |
|         |                              | 2.52 |                                                                                                                        | $\pm 3\%$ @ 90 to 100% humidity                                                                  |
|         | Met One Wind Speed Sensor    | 2.53 | $0-60 \text{ m s}^{-1}$                                                                                                | ± 0.11 m s-1 or 1.5% FS                                                        |
|         | Snow Depth Sensor            | 1.00 | $-45 \text{ to } +50^{\circ}\text{C}$                                                                                  | Maximum of $\pm 1$ cm or 0.4% of distance                                                        |
|         | (SR50A)                      |      |                                                                                                                        | to target                                                                                        |
| Stable  |                              |      |                                                                                                                        |                                                                                                  |
| Plateau |                              |      |                                                                                                                        |                                                                                                  |
| Tripod  |                              |      |                                                                                                                        |                                                                                                  |
|         | Four Component               | 2.00 | 0.3-50 μm                                                                                                       | ± 10%                                                                                     |
|         | Kadiometer (UNKI)            |      |                                                                                                                        | Terms 1 0 200 @ 2000                                                                             |
|         | Temperature and Relative     | 2.00 | Temp: -40 to +60°C
RH: 0-100%                                                                         | $\frac{1 \text{ emp: } \pm 0.2^{\circ} \text{ C } @ 20^{\circ} \text{ C}}{20^{\circ} \text{ C}}$ |
|         | Humidity Probe (HMP45C)      | 2.00 |                                                                                                                        | KH: <math>\pm 2\%</math> @ 0 to 90% humidity                                              |
|         |                              |      |                                                                                                                        | $\pm$ 3% @ 90 to 100% humidity                                                                   |

**Thawing**

Plateau

**Tripod**

| Four Component
Radiometer (CNR1)                 | 1.85                                     | 0.3-50 µm                               | ±10%                                                                                      |
|-----------------------------------------------------|------------------------------------------|------------------------------------------------|--------------------------------------------------------------------------------------------------|
| Temperature and Relative
Humidity Probe (HMP45C) | 2.15                                     | Temp: -40 to +60°C
RH: 0-100% | Temp: ± 0.2°C @ 20°C
RH: ± 2% @ 0 to 90% humidity
± 3% @ 90 to 100% humidity |
| Snow Depth Sensor
(SR50A)                        | 1.43                                     | -45 to +50°C                                   | $\frac{\text{Maximum of } \pm 1 \text{ cm or } 0.4\% \text{ of distance}}{\text{to target}}$     |
| Ground Heat Flux Plate
(HFT3)                    | - 0.05 m                                 | $-40 \text{ to } +55^{\circ}\text{C}$          | better than $\pm$ 5% of reading                                                                  |
|                                                     | 0.0 m                                    |                                                | $\frac{\text{Worst case:}}{\pm 0.4^{\circ}\text{C} (-24 \text{ to } 48^{\circ}\text{C})}$        |
|                                                     | - 0.05 m
- 0.1 m                      |                                                | ± 0.9°C (-35 to 50°C)                                                                     |
|                                                     | - 0.15 m
- 0.2 m                      |                                                | $\frac{\text{Interchangeability Error:}}{\pm 0.10^{\circ}\text{C (0 to 50^{\circ}\text{C})}}$    |
| Thermistors (10/B)                                  | - 0.25 m
- 0.3 m                      | -35 to +50°C                            | $\pm 0.20^{\circ}C \text{ at } -10^{\circ}C$ $\pm 0.30^{\circ}C \text{ at } -20^{\circ}C$        |
|                                                     | - 0.4 m
- 0.5 m                       |                                                | $\pm 0.40^{\circ}$ C at $-30^{\circ}$ C
$\pm 0.50^{\circ}$ C at $-40^{\circ}$ C               |
|                                                     | - 0.7 m                                  |                                                | Steinhart-Hart
Equation Error: $\leq \pm 0.01^{\circ}$ C (-35 to 50°C)                 |
| Water Content
Reflectometers (CS615)             | - 0.1 m
- 0.2 m
- 0.3 m
- 0.4 m | N/A (site specific)                     | $\pm 2\%$ when using calibration for certain
soil                                             |
|                                                     | - 0.5 m                                  |                                                |                                                                                                  |